# A Novel Ensemble Fault Diagnosis Model for Main Circulation Pumps of Converter Valves in VSC-HVDC Transmission Systems

**DOI:** 10.3390/s23115082

**Published:** 2023-05-25

**Authors:** Sihan Zhou, Liang Qin, Yong Yang, Zheng Wei, Jialong Wang, Jing Wang, Jiangjun Ruan, Xu Tang, Xiaole Wang, Kaipei Liu

**Affiliations:** 1Hubei Key Laboratory of Power Equipment & System Security for Integrated Energy, Wuhan 430072, China; 2School of Electrical Engineering and Automation, Wuhan University, Wuhan 430072, China; 3State Grid Economic Technology Research Institute Co., Ltd., Beijing 102200, China

**Keywords:** fault diagnosis, VSG-HVDC, converter station, converter valve, main circulation bump, ensemble learning, machine learning

## Abstract

The intelligent fault diagnosis of main circulation pumps is crucial for ensuring their safe and stable operation. However, limited research has been conducted on this topic, and applying existing fault diagnosis methods designed for other equipment may not yield optimal results when directly used for main circulation pump fault diagnosis. To address this issue, we propose a novel ensemble fault diagnosis model for the main circulation pumps of converter valves in voltage source converter-based high voltage direct current transmission (VSG-HVDC) systems. The proposed model employs a set of base learners already able to achieve satisfying fault diagnosis performance and a weighting model based on deep reinforcement learning that synthesizes the outputs of these base learners and assigns different weights to obtain the final fault diagnosis results. The experimental results demonstrate that the proposed model outperforms alternative approaches, achieving an accuracy of 95.00% and an *F*_1_ score of 90.48%. Compared to the widely used long and short-term memory artificial neural network (LSTM), the proposed model exhibits improvements of 4.06% in accuracy and 7.85% in *F*_1_ score. Furthermore, it surpasses the latest existing ensemble model based on the improved sparrow algorithm, with enhancements of 1.56% in accuracy and 2.91% in *F*_1_ score. This work presents a data-driven tool with high accuracy for the fault diagnosis of main circulation pumps, which plays a critical role in maintaining the operational stability of VSG-HVDC systems and satisfying the unmanned requirements of offshore flexible platform cooling systems.

## 1. Introduction

In recent years, there has been a surge in global electricity consumption, emphasizing the significance of voltage source converter-based high-voltage direct current transmission (VSG-HVDC) systems, especially those based on offshore platforms that have garnered increasing attention [1]. The converter valve is a crucial piece of equipment in the VSG-HVDC system, operating with high power and generating substantial heat, which causes a sharp rise in the temperature of key components such as reactors and thyristors. Failure to cool the valve body promptly can directly affect the electrical characteristics and service life of the converter valve and can even cause the entire transmission system to malfunction [2]. Therefore, the normal operation of the cooling system of the converter valve is critical for maintaining the safe and stable operation of the power system. Among the cooling system components, the main circulation pump is the only equipment that continuously maintains operation by ensuring constant pressure of the internal water-cooled cooling medium and providing continuous medium flow [3]. Any malfunction in the main cycle can directly affect the cooling effect of the converter valve, which may even force the shutdown of the power system [4]. Therefore, it is essential to monitor the operational faults of the main circulation pump in real time so as to avoid further damage to the VSC-HVDC transmission system by promptly repairing any malfunctions.

In the field of fault diagnosis for various power devices, including the main circulation pump, two primary methods are available. The first approach entails regular disassembly and inspection of the main circulation pump by operation and maintenance personnel. However, this method is economically impractical due to the substantial allocation of manpower and time it necessitates as well as the mandatory equipment shutdown [4]. The second approach involves intelligent data-driven fault diagnosis based on operational data. In recent years, the demand for intelligent fault diagnosis of critical equipment within the cooling system of converter valves, notably the main circulation pump, has intensified due to several compelling reasons. Firstly, such a system can alleviate the workload burden on human operators. Moreover, continuous manual monitoring and judgment are unfeasible throughout the entire day, which can lead to undetected faults within the monitoring window, potentially resulting in severe consequences. Furthermore, there has been an unprecedented upsurge in the construction and application prospects of offshore wind power platforms. Given their remote locations and extended unmanned operational periods, the necessity for intelligent fault detection models has become increasingly urgent.

Currently, there are a few mature technical solutions to address the challenges posed by the unmanned operation and maintenance of the main circulation pump in the cooling system of the VSG-HVDC system [5,6,7,8,9,10,11,12,13]. Specifically, a relevant study focused on estimating the health state of the pump [5]. Another research effort involved the development of mathematical models to analyze the electromagnetic and electromechanical processes of the pump during startup and operation modes [7]. In a separate investigation, the impact of the pump speed on the cooling water flow rate was examined to enhance the system’s cooling efficiency [8]. Furthermore, an optimized pump structure was proposed to minimize energy consumption [11]. While these studies shed light on the physical mechanism of the main circulation pump, there is a lack of implementation of specific tasks related to fault diagnosis. In [13], a novel approach to predicting the overtemperature state of some core devices of converter valves, including the main circulation pumps, was proposed. However, this study only focused on the fault of overtemperature without taking into consideration of other fault modes. Given the research value of the operational data generated during the main circulation pump’s operation, it is essential to develop data-driven methods for fault detection of the main circulation pump.

As the field of artificial intelligence (AI) continues to expand, an increasing number of algorithms and models are being utilized for the fault diagnosis of various devices. For instance, a multi-layer perceptron (MLP) model was proposed in [14] for the classification of characteristic signals and the diagnosis of gear faults, demonstrating relatively high stability to signal noise and disturbance. Reference [15] combined support vector machine (SVM) and fuzzy control methods to diagnose faults in steam turbines with high accuracy, and this method comes with high robustness due to the simplicity of hyperparameter settings. In [16], the possibility of applying random forest (RF) for the fault diagnosis of induction motors was investigated, establishing multiple decision trees to improve the single tree classifier, achieving higher recognition speed and precision. Reference [17] proposed a method for the autonomous recognition of alarm events using a convolutional neural network (CNN) for the local feature extraction of alarm information to achieve intelligent recognition of alarm events. In [18], the long and short-term memory artificial neural network (LSTM) is combined with an automatic fault feature extraction method, performing well in rolling bearing condition monitoring and fault diagnosis tasks despite fewer samples and more features. Reference [19] utilizes a fault diagnosis model based on gated recursive unit (GRU), which demonstrates good generalization ability in different fault modes of rolling bearings in rotating machinery. While these studies have explored the potential of various algorithms or models in the field of fault diagnosis, it has been found that using any of these algorithms alone for fault diagnosis of the main circulation pump cannot achieve adequate data feature mining in complex operating conditions and with few fault samples. Therefore, an ensemble learning model has been proposed to address this problem by integrating multiple algorithms.

The most critical aspect of ensemble learning is how to determine the final output of the entire model based on the results of each base learner [20], namely to determine the weights for all base learners. One approach is to take the average of the results output by all base learners as the final output, as adopted in [21]. However, this method is only suitable for regression tasks and not classification tasks, as taking the average of the numbers representing fault labels has no clear mathematical meaning. References [22,23] proposed the plurality voting method and the major voting method, respectively, to determine the recognized fault type based on the majority vote of the base learners. However, these methods assume that the common result recognized by most base learners is the optimal result without considering the situation where only a few base learners perform well under specific conditions. Therefore, there is still room for improvement in the task of weight assignment. To address this issue, many algorithms have been developed to determine the weight of each base learner’s output in the final output. A typical example was described in [24], which used the improved sparrow algorithm (ISA) to assign weights to each base learner, thus considering the comprehensive performance of all working conditions of the monitored device. However, these optimization algorithms are greatly influenced by hyperparameters, easily trapped in local optima, and require the careful selection of basic learners to use.

To address the limitations of current fault diagnosis models for the main circulation pump in the cooling system of a VSG-HVDC converter valve, this study proposes a novel and cost-effective approach that integrates a set of base learners with a weighting model based on deep reinforcement learning (DRL). This approach aims to achieve more accurate fault diagnosis and thus improve the safety and stability of the VSG-HVDC transmission system under unmanned conditions. Each base learner is optimized for specific operating conditions to achieve the best fault diagnosis performance, while the DRL-based weighting model synthesizes their outputs and assigns different weights to obtain the final fault diagnosis results. The contributions and innovations of this study can be summarized in three aspects:(i)This study has conducted a comprehensive validation of different fault diagnosis models for the main circulation pump in the specific application scenario. The findings reveal that while LSTM performs the best among individual models, the ensemble comprising MLP, CNN, and LSTM surpasses both the individual models and other ensembles in terms of diagnostic performance. This highlights the strength of utilizing a combined approach for improved fault diagnosis outcomes.(ii)The contribution of this article also lies in the development of a proposed model that effectively harnesses the strengths of different AI models, thereby overcoming the limitations associated with existing fault diagnosis algorithms when applied to complex operating conditions and limited fault data scenarios in main circulation pumps. Specifically, a novel training approach called the coach-members method is introduced for ensemble models, enhancing the compatibility of diverse base learners across various working conditions and resulting in an ensemble fault diagnosis model that achieves superior performance. This innovative approach demonstrates the capability to overcome challenges associated with fault diagnosis in demanding operational environments.

The remaining part of this study is organized as follows. In Section 2, a brief introduction to the main circulation pump is provided along with a mathematical model for fault diagnosis. Section 3 outlines the proposed framework for the ensemble fault diagnosis model, describes its components, and details the training methods for this model. In Section 4, the effectiveness of the proposed model in the fault diagnosis task of the main circulation pump is validated through three cases with a progressive relationship. Finally, Section 5 summarizes the important conclusions of this study.

## 2. Overview of Fault Diagnosis for the Main Circulation Pump

### 2.1. A Brief Introduction to the Main Circulation Pump

The main circulation pump plays a crucial role in the internal water cooling system, as it is the only rotating equipment that operates continuously. Its normal operation ensures sufficient pressure and maintains a constant flow of the cooling water [25]. The heat exchange process facilitated by the main circulation pump helps to keep the thyristor and its components within the optimal operating temperature range [25]. Therefore, if the main circulation pump experiences any malfunction, it can result in the loss of power to the cooling system, which can ultimately lead to direct current blocking.

The cooling system inside each pole of the converter valve usually has two main circulation pumps, with one serving as the primary pump and the other serving as a backup. The main circulation pump is made up of several components, including a motor, impeller, shaft, bearings, and pump housing [26]. The motor supplies the necessary power to the impeller, which in turn drives the flow of cooling water. The shaft connects the impeller and motor, while the bearings reduce friction and provide support for the shaft. The pump housing encloses the impeller, and the inlet and outlet for the cooling water are located on the housing. Despite its critical role in the cooling system, the main circulation pump is prone to faults. There are three common faults associated with the pump [27]:(i)Bearing lubrication failure: usually caused by insufficient lubrication or the use of low-quality lubricants, leading to increased friction and heat generation in the bearings. This can result in premature failure of the bearings and potential damage to the shaft.(ii)Shaft center deviation: usually caused by improper assembly or installation, resulting in misalignment between the shaft and bearings. The consequences of shaft center deviation include increased vibration, wear and tear on the bearings, and potential damage to the impeller.(iii)Water leakage: due to wear and tear on the pump housing or impeller or inadequate sealing of the inlet or outlet. The consequences of water leakage include reduced coolant flow, increased pressure in the cooling system, and potential damage to other components in the system.

Therefore, it is crucial to monitor the main circulation pump for common faults to prevent potential damage to the system. Each main circulation pump is equipped with four vibration sensors that monitor horizontal and vertical vibration on the pump and motor sides, and the collected vibration signals can be used to diagnose the faults of the main circulation pump. However, the performance of different methods to process these data for fault diagnosis varies, and existing commonly used methods often fall short in accurately diagnosing main circulation pump faults. Hence, it is of paramount importance to enhance existing methods and develop a new and effective fault diagnosis model.

### 2.2. The Mathematical Model for Main Circulation Pump Fault Diagnosis

In order to diagnose the fault of the main circulation pump, sufficient time series data should be used as input to the fault diagnosis model, as shown in Equation (1):(1)$$X_{t}=[x_{t},x_{t+1},\cdots ,x_{t+q}]$$
where *q* is a hyperparameter which represents the length of the data sequence containing enough information and is to be determined in Section 4.1. $x_{t}$ is the data sequence sampled at time t, which is composed of signals from different sources and can be expressed as
(2)$$x_{t}={\left[x_{t}^{1},x_{t}^{2},\cdots ,x_{t}^{m}\right]}^{T}$$
where m is the number of signals sampled at one timestamp. According to Section 2.1, there are four vibration signals sampled in the fault diagnosis task of the main circulation pump, so m is set to be 4.

The main circulation pump of the VSG-HVDC system has three common faults in addition to its normal operating mode, namely: bearing lubrication failure, shaft center deviation, and water leakage. The normal working mode is marked as 0, while the remaining three faults are marked as 1, 2 and 3, respectively. $y_{t}$ is used to represent the fault situation at the current time, as is shown in Equation (3):(3)$$y_{t}\in \left\{0,1,2,3\right\}$$

Many methods can be used for fault diagnosis of the main circulation pump in the VSG-HVDC system. By inputting the collected data $X_{t}$, the current fault type can be predicted, as shown in Equation (4):(4)$${\hat{y}}_{t}=f(X_{t})$$
where ${\hat{y}}_{t}$ is the predicted fault type of the current timestamp and f(·) is the implicit function of the fault diagnosis model.

However, due to the complex operating conditions of the main circulation pump, it is difficult for a single model to accurately fit the relationship between operating data and faults. Therefore, we consider using multiple popular fault diagnosis models (referred to as base learners) simultaneously to jointly diagnose faults. The fault diagnosis result of one of the base learners is:(5)$${\hat{y}}_{t}^{n}=f_{i}(X_{t})$$
where n=1,2,⋯,p is the index of base learners, and p refers to the number of base learners, which is tentatively set to 3. When a new base learner is introduced or a base learner that is currently in use needs to be removed, p will also be modified accordingly.

After all the base learners have output the results of fault diagnosis, a model is required to coordinate these outputs. This model that plays a coordinating role needs to comprehensively consider the current operating conditions and the conclusions given by each model in order to output a reliable diagnostic result, as shown in Equation (6):(6)$${\hat{y}}_{t}=h(X_{t},{\hat{y}}_{t})$$
where h(·) is the implicit function of the coordinating model and ${\hat{y}}_{t}=[{\hat{y}}_{t}^{1},{\hat{y}}_{t}^{2},\ldots ,{\hat{y}}_{t}^{p}]$ is the vector of diagnosis results. Different coordinating models are characterized by different implicit functions h(·), and in this study, Equation (7) can quantify the implicit function h(·) that coordinates the diagnostic results of each base learner:(7)$${\hat{y}}_{t}={\hat{y}}_{t}\cdot w_{t}^{T}=\sum_{i=1}^{p}{\hat{y}}_{t}^{n}\cdot w_{t}^{n}$$
where $w_{t}$ represents the weight vector for base learners and is obtained by the DRL model and $w_{t}^{n}$ is its No. *n* element. The diagram of coordination of diagnosis results of different base learners is present in Figure 1.

In the field of fault diagnosis, four performance metrics, namely accuracy, precision, recall, and *F*_1_ score, are often used to evaluate the performance of fault diagnosis models [28]. As there are four operating modes, including one normal mode and three fault modes, it is necessary to calculate these four performance metrics separately for each operating mode and average the values of each metric across the four categories. For each operating mode, data diagnosed as the given mode are marked as positive (P), which can be further divided into true positive (TP) and false positive (FP), representing the amount of data that actually belong to and do not belong to the given mode, respectively. The other three categories that do not belong to this category are labeled negative (N), which can be similarly divided into true negative (TN) and false negative (FN), representing the data that belong to and do not belong to those categories outside of the given category in the real situation. The number of TP, FP, TN, and FN under each classification will be used in the calculation of the following performance metrics:(i)Accuracy (acc) is used to measure the overall correctness of the fault diagnosis model, which is defined as follows:
(8)$$acc=\frac{TP+TN}{TP+FP+TN+FN}$$

(ii)Precision (pre) aims to determine the proportion of correctly diagnosed positive samples, which can be calculated using Equation (8):


(9)
$$pre=\frac{TP}{TP+FP}$$


(iii)Recall rate (rec) aims to determine the proportion of actual positive samples that are correctly predicted as positive, and it can be calculated using the following equation:


(10)
$$rec=\frac{TP}{TP+FN}$$


(iv)*F*_1_ score ($F_{1}$) is a metric that reflects both accuracy and recall. It is defined as follows:


(11)
$$F_{1}=\frac{2\times Pre\times Rec}{Pre+Rec}$$


## 3. Model Construction

### 3.1. Base Learners for Fault Diagnosis

In recent years, AI methods have gained increasing attention for their potential to improve the accuracy and efficiency of fault diagnosis. Among the numerous AI methods, MLP, CNN, and LSTM have emerged as popular choices for fault diagnosis tasks due to their ability to capture complex patterns, handle noisy data, and capture temporal dependencies. A brief overview of these three algorithms is provided as follows, highlighting their professional fundamentals and advantages in fault diagnosis.

(i)MLP: a feedforward neural network model that consists of multiple layers of interconnected nodes and can be trained using backpropagation. It is known for its ability to learn complex non-linear relationships between inputs and outputs, making it suitable for capturing patterns in fault data. The advantages of MLP in fault diagnosis include its flexibility in handling different types of data, ability to learn complex patterns, and potential for accurate classification and prediction. However, MLP may have limitations in handling noisy data, overfitting, and sensitivity to hyperparameter tuning.(ii)CNN: a type of deep neural network that is designed for image or data in the form of an array. It uses convolutional layers to automatically learn spatial hierarchies of features from data, making it highly suitable for fault diagnosis tasks where the fault data are presented as an array. CNN has been shown to achieve state-of-the-art performance in many fault diagnosis tasks, making it a popular choice in the field.(iii)LSTM: a type of recurrent neural network (RNN) that is designed to overcome the vanishing gradient problem in traditional RNNs, making it more effective in capturing long-term dependencies in sequential data. LSTM introduces specialized memory cells with gating mechanisms that can control the flow of information, allowing for the effective handling of sequences with varying time scales. LSTM has gained popularity in fault diagnosis tasks that involve time series data due to its ability to capture complex temporal dependencies, handle variable-length sequences, and mitigate the vanishing gradient problem.

Despite the advantages of each individual model, no single model is perfect and can perform well in all operational conditions of the VSG-HVDC system’s main circulation pump, since each model has its own limitations and may struggle with certain types of faults, noisy data, or complex data distributions. Therefore, a practical approach is to ensemble these models to leverage their strengths and mitigate their weaknesses.

The ensemble approach in fault diagnosis has several advantages. First, it can reduce the risk of misclassification by leveraging the strengths of different models. Second, the ensemble approach can enhance the generalization performance of the models, as they can compensate for each other’s limitations. Finally, the ensemble approach can provide a more reliable diagnosis result by considering the consensus of multiple models. In case one model produces an incorrect diagnosis due to its limitations or noise in the data, the ensemble can still make a more accurate decision based on the majority vote or weighted average of the models’ predictions.

### 3.2. The DRL-Based Weighting Model

To integrate the diagnostic results of multiple base learners, it is necessary to determine the degree to which their outputs should be weighted, which is a critical aspect of ensemble learning-based fault diagnosis. Various methods have been proposed to process the results of multiple base learners, including the average weighting method [21] and the voting method [22,23], which are based on taking the average or the majority vote of the outputs. Although ISAs have been proposed to address the limitations of these methods [24], they do not consider the variation in the fault diagnosis performance of different base learners under different operating conditions. Empirical testing shows that the accuracy of these methods can still be improved, which is to be discussed in Section 4.3. To address this issue, we propose a novel method for determining weights based on DRL, which is an algorithmic framework that models the mapping between environmental states and actions with the ultimate objective of maximizing the cumulative reward that an agent receives through iterative trial-and-error interactions with a given environment [29]. This approach leverages the powerful perception and decision-making capabilities of DRL to assign weights to each base learner based on the operating conditions of the main circulation pump, which enables the integration of the diagnostic results of multiple base learners to achieve more accurate fault diagnosis.

The basis for determining $w_{t}$ is the operating condition of the main circulation pump and the output result of each base learner, characterized by $X_{t}$ and ${\hat{y}}_{t}$, respectively. Hence, the state variable $s_{t}$ for DRL is defined as follows:(12)$$s_{t}=[X_{t},{\hat{y}}_{t}]$$

DRL is employed to assign the weights for all base learners, so the action variable $a_{t}$ is equivalent to the weight vector, as shown in Equation (13):(13)$$a_{t}=w_{t}=[w_{t}^{1},w_{t}^{2},\cdots ,w_{t}^{p}]$$

The aim of utilizing the DRL-based weighting model is to obtain the optimal weight combination at each time point, thereby minimizing the average error (or cumulative error) of the fault diagnosis results of the ensemble model. This objective can be transformed into a task of maximizing cumulative rewards within the DRL framework. Accordingly, the agent’s reward function $r_{t}$ can be expressed as:(14)$$r_{t}=\left\{\begin{array}{cc} 0 & ,if\;{\hat{y}}_{t}\in \left\{y_{t-q},y_{t-q+1},\cdots ,y_{t},y_{t+1},\cdots ,y_{t+q}\right\}\\ -{\left({\hat{y}}_{t}-y_{t}\right)}^{2}/2 & ,if\;{\hat{y}}_{t}\notin \left\{y_{t-q},y_{t-q+1},\cdots ,y_{t},y_{t+1},\cdots ,y_{t+q}\right\} \end{array}\right.$$

It is important to note that the start and end times of faults in the main circulation pump operating data can be difficult for operation and maintenance personnel to define accurately. In reality, faults may last for a certain period of time. Therefore, when comparing the fault diagnosis results of the ensemble learning model at a particular timestamp to the actual results, it is necessary to consider the practical results that match the output of the ensemble learning model within a range of q steps forward and backward of the timestamp. If a consistent result is found, the diagnosis is considered correct and $r_{t}=0$, resulting in no change in the cumulative reward. Conversely, if the diagnostic result is incorrect, $r_{t}$ will be negative, causing the cumulative reward to decrease.

### 3.3. The Coach-Members Method: Training of the Ensemble Fault Diagnosis Model

The proposed ensemble fault diagnosis model is a highly parameterized model, with hyperparameters that require careful selection and a strict training process. This model includes p base learners and a DRL-based weighting model. To ensure the correct setting of hyperparameters, each of the component models should initially be equipped with multiple hyperparameter combinations to generate a sufficient number of candidate models. The optimal hyperparameter combination for each component model should be selected from the candidate models using a validation set. The base learners require two stages of training, namely, rough training and fine training, and the coach-members method is employed in the latter stage. All base learners should undergo both stages of training while the DRL-based weighting model only requires the second stage. The application process of this ensemble fault diagnosis model is illustrated in Figure 1, and its training process is shown in Figure 2.

The specific steps of obtaining the optimal ensemble fault diagnosis model are shown as follows:(i)Normalize the sampled vibration signal data by mapping it into [0,1]. The normalization formula is in shown in Equation (15):
(15)$$x\to x^{N}=\frac{x-x_{\min}}{x_{\max}-x_{\min}}$$
where x is the original sample data, and $x^{N}$ is its normalized form; $x_{max}$ and $x_{min}$ represent the maximum and minimum values of the sample dataset, respectively.

(ii)Divide the sample dataset into a training set, validation set and test set by the ratio of 3:1:1.(iii)For each base learner, obtain candidate models with different combinations of hyperparameters using manual experience and grid search methods.(iv)Perform the first-stage rough training for all candidate base learners on the training set.(v)Evaluate the trained candidate base learners on the validation set. From each type of candidate base learner, select the optimal one producing the lowest error, respectively, so as to form the combination of p roughly trained base learners.(vi)Similarly, obtain candidate DRL-based weighting models with different combinations of hyperparameters. Obtain the same number of candidate ensemble fault diagnosis models by coordinating each candidate DRL-based weighting model with the roughly trained base learner combination, respectively.(vii)Perform the second-stage fine training for all candidate ensemble fault diagnosis models on the training set.(viii)Evaluate the trained candidate ensemble fault diagnosis models on the validation set and select the optimal one producing the lowest error.(ix)Test the optimal ensemble fault diagnosis model on the test set. Denormalize and analyze the output with the performance metrics, as shown in (6) to (9).

Stage (vii) employs the coach-members method proposed in this study, which is inspired by the concept of training football teams. To win a game, players not only need to meet their own level standards but also require an experienced coach to develop appropriate strategies for them based on the situation on the field and the actual condition of the players. The coach needs to arrange the optimal roles and participation levels of each member in the game, and each member needs to improve their cooperation with other players under the coach’s guidance, thereby increasing the probability of winning. If a player has strong personal abilities but receives poor guidance from the coach, or if the coach has high coordination skills but each player’s personal level or ability to cooperate with others is poor, the team’s probability of winning will be very low. Therefore, to improve the team’s probability of winning, the following three conditions need to be met simultaneously:(i)When members join the team, they should already have a good foundation in terms of their abilities and skills;(ii)The coach is able to optimize the team’s cooperation strategies in real time based on the situation on the field and the actual capabilities of each member;(iii)Members are able to optimize their collaboration with each other in real time based on the strategies provided by the coach, allowing for efficient communication and synergy among team members.

The methods mentioned above to improve the winning probability can be summarized as the coach-members method. Similarly, to enhance the performance of the proposed ensemble fault diagnosis model in the task of detecting faults in the main circulation pump, the model needs to meet the following three conditions:(i)Before being incorporated as a base learner in the ensemble fault diagnosis model, each fault diagnosis model must have its hyperparameters and parameters fine-tuned to perform the fault diagnosis task independently and with satisfactory performance.(ii)The DRL-based weighting model should continuously optimize its parameters based on diagnostic errors. This allows it to determine the weight of each base learner more appropriately based on the operating status of the main circulation pump and the output of each base learner. As a result, each base learner can leverage its strengths and avoid weaknesses in diagnostic tasks under different working conditions.(iii)All base learners should continuously and simultaneously optimize their respective parameters based on diagnostic errors, so that when the DRL-based weighting model provides specific weight combinations, they can improve their collaboration with other base learners and enhance the accuracy of fault diagnosis.

The first condition of having each base learner handle the fault diagnosis task independently with good performance is satisfied after the first-stage rough training. However, the second and third conditions require implementation through the second stage of fine training. During each batch of the training process, the weight values are given by the weighting model, the predicted value of $y_{t}$ is calculated, the prediction error is obtained, and the DRL parameters are updated using gradient descent; next, all base learners provide diagnostic results, and the parameters of base learners are optimized using backpropagation until all the data in the training set are utilized, marking the completion of the second stage of fine training.

It is important to note that in the second stage of fine training, all base learners must be involved. Although the base learners that have undergone the first stage of rough training have a certain level of accuracy in fault identification, this accuracy represents the average level across all operating conditions. A good ensemble fault diagnosis model should not solely rely on the overall performance of each base learner, as this may result in common shortcomings in fault diagnosis tasks for all base learners under certain operating conditions. Instead, a good ensemble fault diagnosis model should require all base learners to coordinate and cooperate, so that some base learners have high accuracy in each working condition, and the outputs of these high-accuracy base learners will have a greater influence on the final output, thus improving the overall performance of fault diagnosis.

## 4. Case Study

In this section, we demonstrated the excellent performance of the proposed model in the fault diagnosis task of the main circulation pump of the converter valve in the VSG-HVDC system through three progressive cases. Case 1 examined the performance of five commonly used fault diagnosis models and obtained their hyperparameter configuration and initial network parameters. In Case 2, we found the optimal combination of base learners that maximized the improvement in performance metrics. Building upon the optimal base learner combination found in Case 2, Case 3 verifies the superior performance of the DRL-based weighting model proposed in this study compared to other methods.

### 4.1. Data Source and Division

The experiment used data collected from the main circulation pump data collection system of the No.1 converter valve at Chuxiong converter station in China, from 00:00 a.m. on 1 January 2018 to 20:00 p.m. on 31 December 2020. The main parameters of the main circulation pump include its model, CPKN C200-500, a flow rate of 107 L/s, lift of 65 m, speed of 1450 rpm, and rated power of 110 kW. Four vibration signals were collected, namely the pump side horizontal vibration signal, motor side horizontal vibration signal, pump side vertical vibration signal, and motor side vertical vibration signal, with a sensor sampling frequency of 2 kHz.

Before using the collected data in this experiment, some preprocessing is required. First, the three-year dataset must be divided into several independent and sufficiently long sequences of data to be processed by the model. The data partitioning follows three principles:(i)During routine maintenance, the converter valve was shut down nine times from 2018 to 2020, resulting in the main circulation pump’s vibration signal data being divided into roughly 10 sections by the nine breakpoints.(ii)Each data point is labeled by the operation and maintenance personnel of the converter station with a certain type of fault. To reduce the information extraction burden of the model, independent data sequences should only contain data points with the same fault type.(iii)The selection of the length of each independent data sequence, i.e., q in (1), requires rigorous calculation. The principle of calculation is to minimize the complexity of training while ensuring the integrity of the temporal information contained in each data sequence. Trappenberg et al. [30] proposed a method for calculating the optimal q based on the principles mentioned above, and in this study, q is set to 2380 using this method.

Based on the above partitioning principles and a 3:1:1 ratio, the dataset was divided into a training set, a validation set, and a test set. The training set contained 240 data sequences for each of the four fault modes, while the validation and test sets had 80 data sequences for each of the four fault modes, respectively.

### 4.2. Case 1: Evaluation for Single Base Learners

This case is used to evaluate whether some algorithms are suitable to be used as the base learners of the ensemble fault diagnosis model and also to obtain the optimal hyperparameter settings of these base learners. In addition to MLP, CNN, and LSTM introduced in Section 3.1, RF and GRU were also tested and indexed as shown in Table 1.

After the first-stage rough training described in Section 3.3, the important hyperparameters of these models are shown in Table A1. The trained models were then tested using a test set, and their performance was evaluated using the four performance metrics introduced in Section 2.2. The results are shown in Table 1.

Based on the results in Table 1, it can be concluded that all six models achieved acceptable performance to varying degrees, thus validating the previous studies [14,15,16,17,18,19]. This indicates that these models can serve as base learners for the ensemble fault diagnosis model. Although LSTM outperformed the other models in terms of accuracy, MLP performed the worst. However, considering the four performance metrics, there is still significant room for improvement, and more effective methods will be further evaluated in Cases 2 and 3.

### 4.3. Case 2: Evaluation for Ensemble Learning Using Different Combinations of Base Learners

In this case, we selected different numbers of models as base learners to create a combination of base learners, and then, we employed the DRL-based weighting model using the coach members method to assign weights to the diagnostic results of each base learner. From Case 1, where there were a total of five tested models, we selected two, three, four, and five models as base learners to create a combination of base learners, resulting in 10, 10, 5, and 1 ensemble fault diagnosis models, respectively, for a total of 26. The index of these combinations of base learners and the base learners included within them is shown in Table A2.

The trained ensemble fault diagnosis models were tested using a test set, and their performance in fault diagnosis was measured using accuracy, precision, recall, and *F*_1_ score. The results are shown in Table A2. To find the optimal combination of base learners for the main circulation pump fault diagnosis, we calculated the degree of improvement for each combination of base learners based on the performance of MLP in Case 1, using Equation (16), as shown below:(16)$$imp(i)=\sum_{k=1}^{4}\left[{metric}^{k}(i)-{metric}^{k}(MLP)\right]$$
where i represents the No. *i* combination of base learners in Table A2, imp(i) is its performance improvement compared with MLP, k is the index of performance metrics and when k is set to 1, 2, 3 and 4, ${metric}^{k}=acc,pre,rec,F_{1}$, respectively. The improvements of different base learner combinations in main circulation pump fault diagnosis is shown in Figure 3.

Figure 3 shows that the No.14 combination of base learners, which includes MLP, CNN, and LSTM, achieves the best performance in fault diagnosis. Although in Case 1, these three base learners were not the top performers individually, their combination significantly improved the fault diagnosis model’s overall performance. This suggests that these three base learners have achieved optimal coordination and demonstrated good comprehensive performance across all working conditions. When only two base learners were used, the performance improvement was not significant enough, as the combined use of two models was insufficient to fully mine the operating data of the main circulation pump. Moreover, including more than three base learners did not improve performance as much as using {MLP, CNN, LSTM}, in that the addition of other redundant base learners reduced the weight of these three highly fit base learners, resulting in the final output containing some invalid information.

### 4.4. Case 3: Evaluation for Ensemble Learning Using Different Weight Determination Method

To validate the effectiveness of the DRL-based weighting model proposed in Section 3.2 and the coach-members method introduced in Section 3.3, this case employed five different weight determination methods to determine the weights of {MLP, CNN, LSTM} as the optimal base learner combination, as shown in Table 2. The first three methods are the average weighting method employed in [21], the voting method utilized in [22,23], and the ISA-based weighting method proposed in [24], which have been introduced in Section 3.2. The fourth method is the DRL-based weighting model without utilizing the coach-members method for training, which means that the network parameters of these base learners are consistent with the training results of Case 1. The fifth method is the DRL-based weighting model proposed in this article, which employs the coach-members method for model training. The performance of the affordable fault diagnosis models using these five weight determination methods is shown in Table 2 and Figure 4.

From Table 2 and Figure 4, it is evident that the DRL-based weighting model proposed in this article outperforms traditional methods, namely the voting method and the average weighting model. Additionally, utilizing the coach-members method during the training process leads to a further improvement in accuracy and *F*_1_ by 2.18% and 5.27%, respectively. In contrast, if the coach members method was not utilized during training, the model did not perform as well as the ISA-based weighting method. However, when the coach-members method was employed for model training, the model outperformed all other weight determination methods, demonstrating the effectiveness of the proposed DRL-based weighting model.

## 5. Conclusions

This study presents a novel ensemble fault diagnosis model for the main circulation pumps of converter valves in VSG-HVDC transmission systems. The model integrates three base learners, namely MLP, CNN, and LSTM, with a DRL-based weighting model that determines the weights of all base learners in the final diagnosis result. This ensemble fault diagnosis model accurately identifies the fault type of the main circulation pump, which is critical for ensuring operational safety and stability in converter stations and contributes to the unmanned operation, maintenance, and monitoring of offshore VSG-HVDC systems.

The study conducts extensive validation work on the application scenario of fault diagnosis for the main circulation pump, leading to the conclusion that LSTM performs the best among individual models. However, when the ensemble fault diagnosis model is applied, the combination of MLP, CNN, and LSTM outperforms individual models. Furthermore, experimental results demonstrate that the proposed model surpasses traditional AI models in terms of accuracy, precision, recall, and *F*_1_ score. Notably, the four performance metrics reach as high as 95.00%, 95.00%, 86.36% and 90.48%, respectively, which are 4.06%, 8.75%, 7.05% and 7.85% higher than those achieved by the widely used LSTM model, respectively. This highlights that the proposed model effectively leverages the strengths of different models and overcomes the limitations of the existing fault diagnosis algorithms in complex operating conditions with limited fault data. Furthermore, it outperforms the latest existing ensemble model based on the improved sparrow algorithm, with enhancements of 1.56% in accuracy and 2.91% in *F*_1_ score. This indicates that the coach members method enhances the compatibility of different base learners in various working conditions, further enhancing the performance of the ensemble model.

## Figures and Tables

**Figure 1 sensors-23-05082-f001:**
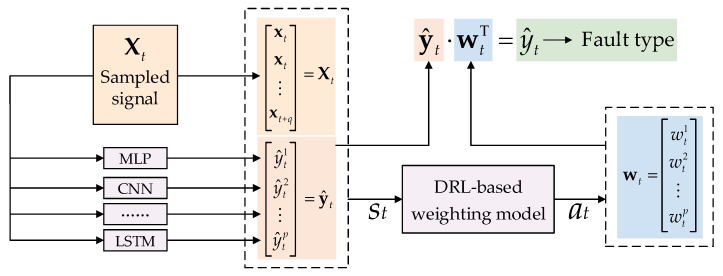
Diagram of coordination of diagnosis results of different base learners.

**Figure 2 sensors-23-05082-f002:**
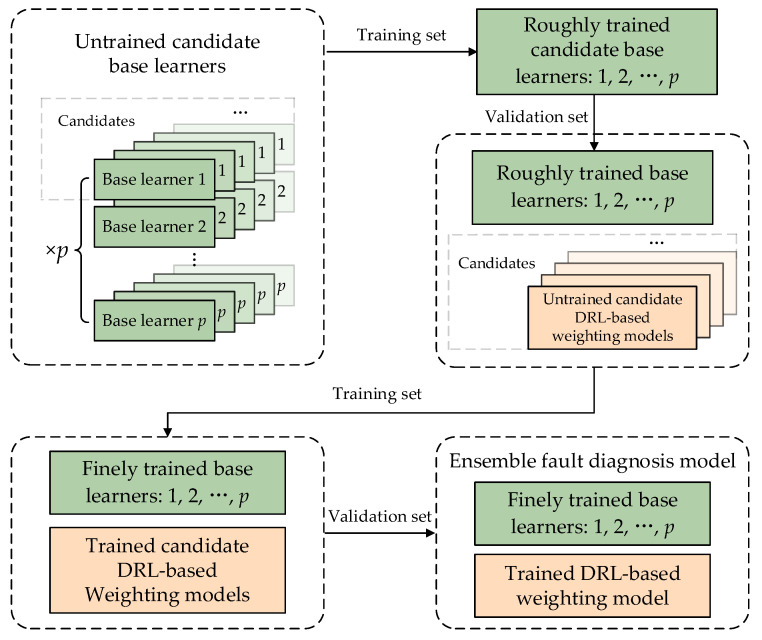
Diagram of model training.

**Figure 3 sensors-23-05082-f003:**
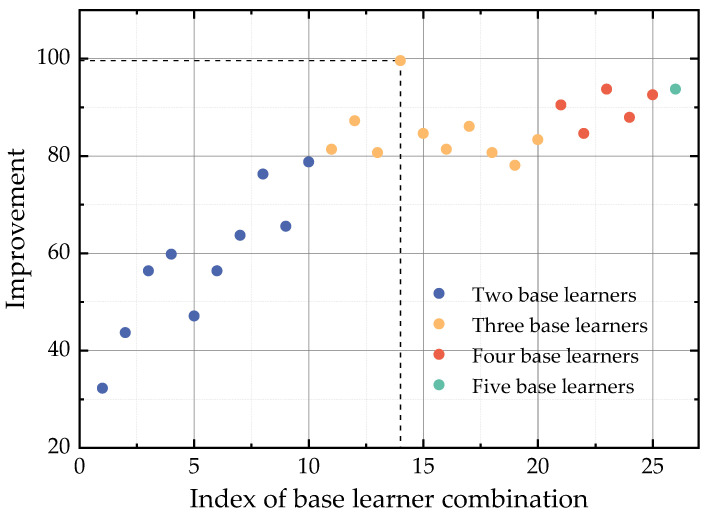
Performance of different base learner combinations.

**Figure 4 sensors-23-05082-f004:**
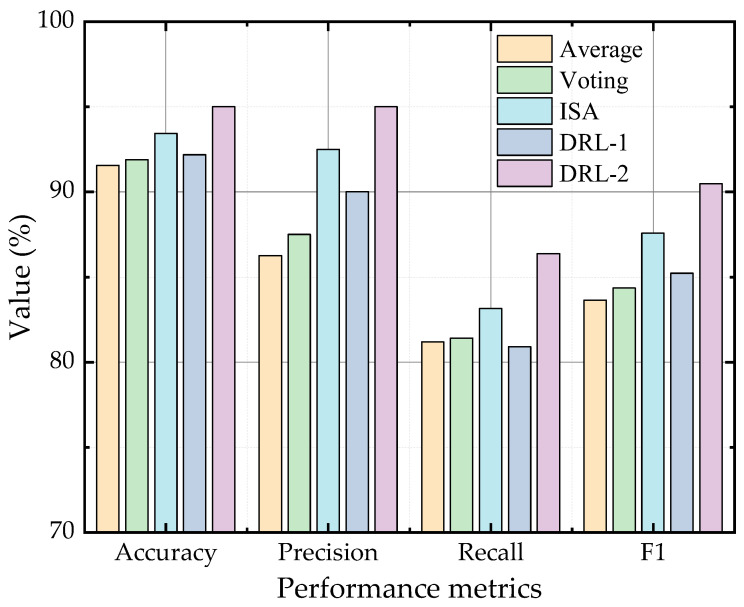
Performance of different weight determination methods.

**Table 1 sensors-23-05082-t001:** Performance of different base learners on the test set.

Model		Accuracy (%)	Precision (%)	Recall (%)	*F*_1_ (%)
Index	Name				
1	MLP	84.06	75.00	65.93	70.18
2	RF	86.56	81.25	69.89	75.14
3	CNN	89.69	86.25	75.82	80.70
4	LSTM	90.94	86.25	79.31	82.63
5	GRU	90.63	87.50	77.78	82.35

**Table 2 sensors-23-05082-t002:** Performance of ensemble fault diagnosis models using different weight determination method.

Method	Accuracy (%)	Precision (%)	Recall (%)	*F*_1_ (%)
Average	91.56	86.25	81.18	83.64
Voting	91.88	87.50	81.40	84.34
ISA	93.44	92.50	83.15	87.57
DRL-1	92.19	90.00	80.90	85.21
DRL-2	95.00	95.00	86.36	90.48

Note: In the first column, “Average”, “Voting”, “ISA”, “DRL-1” and “DRL-2” represent the average weighting method, the voting method, the ISA-based weighting method, the DRL-based weighting model without utilizing the coach-members method for training and the DRL-based weighting model utilizing the coach-members method for training, respectively.

## Data Availability

Not applicable.

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
