# Peer review of "A Novel Ensemble Fault Diagnosis Model for Main Circulation Pumps of Converter Valves in VSC-HVDC Transmission Systems"

_sensors, 2023, doi:10.3390/s23115082_

Round 1

Reviewer 1 Report

1. Methodology is too complex, should be described clearly by a flow/line diagram.

2. What is the role of Xt in Figure 3? Please modify it to make it more clear.

3. Try to adjust the figures, below the text, where they are described.

Author Response

Dear Reviewer,

Thank you for your valuable time and generous time in promoting the quality of this article. We have made careful revisions according to your comments, which we believe can satisfy your expecations. Please find the respond letter in the attachment.

Warmest regards,

Sihan Zhou

Reviewer 2 Report

1- Please enhance the abstract by adding some prominent results.

2- The authors must enhance the Motivation section and discuss more related issues to it.

3- More discription should be added for the methodology

4- The Literature Review section must be updated with recently published papers such as: Improved double‐surface sliding mode observer for flux and speed estimation of induction motors; An IoT-enabled hierarchical decentralized framework for multi-energy microgrids market management in the presence of smart prosumers using a deep learning-based forecaster; An interval-based nested optimization framework for deriving flexibility from smart buildings and electric vehicle fleets in the TSO-DSO coordination; A risk-based bi-level bidding system to manage day-ahead electricity market and scheduling of interconnected microgrids in the presence of smart homes; Energy Management in Microgrids including Smart Homes: A Multi-objective Approach;

5- The contribution section should be written more clearly and highlight the strengths of the paper.

‎‎6- The conclusion section can be updated with prominent and numerical results.

Guys furb

Author Response

(The authors gave the same response as above.)

Reviewer 3 Report

Dear Authors,

I have read your manuscript with great interest. I have some comments on the manuscript:

1. Please revise the sub-section 3.2.1. It has too many similarities with the paper https://doi.org/10.3390/electronics12092075. Including Figures 1-4.

2. The equations in the manuscript should be treated as part of the sentence. There is a lack of punctuation. Please check the requirements of the journal.

3. Approximately 1/3 of references are from the last 5 years. I believe it could be improved. Also please check the requirements of the journal on References.

4. Please check all the abbreviations if they are explained in the text.

5. In the manuscript both "paper" and "study" terms are used. The suggestion would be to have one term.

Author Response

(The authors gave the same response as above.)

Round 2

Reviewer 2 Report

accepted